# BICEC: Attachable Classification-Based Intelligent Control for Sustainable Computer Vision Systems

**Jonathan Burton-Barr [1,2], Deepu Rajan [1], and Basura Fernando[2,1]**
[1]College of Computing and Data Science, Nanyang Technological University, Singapore
[2]Centre for Frontier AI Research, Agency for Science, Technology and Research, Singapore
{BURT0002, ASDRajan}@ntu.edu.sg, Fernando_Basura@cfar.a-star.edu.sg

## Abstract

Computer vision systems can employ multiple vision models to complete a single task or an array of tasks. Reasons may span from no single model being available that meets user requirements, hosting devices lacking the compute to execute a single model that contains the full required functionality, or training a new model requires extensive resources or expertise. Without intelligent input discrimination, these systems risk inefficient processing, leading to increased inference times and energy consumption. This paper investigates the impact of intelligent model activation regulation on energy efficiency and inference speed. We propose BICEC (Branched Image Classification Evaluative Controller), a lightweight solution based on a branched EfficientNetv2 architecture. BICEC can be integrated with existing vision systems with minimal tuning, requiring no retraining of the original system, by creating model-specific branches optimized for minimal size and near-optimal performance. Results show good performance for identifying when a model is relevant and significant reductions in system inference time and energy cost. While the scope of this work focuses on vision systems, we hope to exemplify how tighter control of AI systems can enhance sustainability and computational efficiency.

## 1 Introduction

Intelligent control involves algorithms that optimize system regulation (Åström & McAvoy, 1992). It has proven effective in reducing resource overuse in non-AI systems, examples being optimized thermostat control for heating systems (Nägele et al., 2017) and task scheduling in cloud computing (Rjoub et al., 2021). When we build AI systems composed of multiple models, for a single task or diverse set of tasks, lack of regulation can cause inefficient processing when inputs are processed by irrelevant models. This results in increased energy consumption and computational waste, contributing to the growing concerns over AI's sustainability (Van Wynsberghe, 2021; Thompson et al., 2020). Optimizing model activation can help mitigate these issues and reduce accumulative inference energy costs in AI systems.

Motivation for employing multiple AI models varies by domain, whether to extend system functionality (Yang et al., 2023; Phung et al., 2021), enhance performance (Qayyum et al., 2021; Mogan et al., 2023), or accommodate hardware constraints (Xu et al., 2019; Hu et al., 2019). BICEC (Branched Image Classification Evaluative Controller) proposes input-conscious activation control for vision systems, minimizing unnecessary processing to reduce inference times and energy consumption. This reduction of waste processing can contribute to the development of more sustainable AI systems. BICEC's design aims to be **attachable** and **adaptable**. Attachable, meaning it can be trained independently and integrated without requiring retraining or modification of existing system models. Adaptable meaning it reduces tuning requirements when the attached system changes.

## 2 Methodology

BICEC is an attachable classification-based intelligent controller designed to optimize model activation in computer vision systems. It reduces computational costs and energy consumption by se-

lectively activating relevant models based on input characteristics. BICEC processes inputs through a set of shared layers that learn a common representation for all branches. Each branch corresponds to a specific model and is trained to output a binary classification indicating whether its associated model should be activated. Activation conditions (Table 1) define the features that determine a model's relevance to an input. BICEC undergoes a two-phase training process. **Phase 1:** A base model is established with shared layers and branches derived from a pretrained EfficientNetV2-B0 network (Figure 1a). **Phase 2:** Branches are scaled and optimized to minimize size while maintaining activation accuracy (Figure 1b). During training BICEC is initialized with an input layer per branch, however, after training the input layers for the final model (Figure 1c) are cut to one.

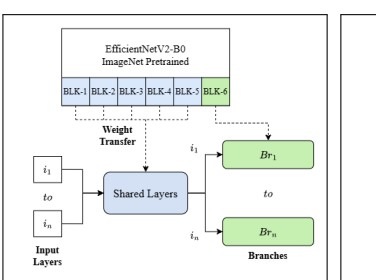

(a) *Phase 1*: Base Creation.

(b) *Phase 2*: Branch Adaptation.

(c) *Final Model*.

Figure 1: Visualization of BICEC during and after training where $P1$ denotes phase 1, $P2$ phase 2, $Br$ branch, $BT$ binary threshold, $i$ input, and $BLK$ denotes an EfficientNetV2 block.

## 2.1 TRAINING PHASE 1: BASE CREATION

**Design.** BICEC is a branched neural network built on shared layers derived from the first five blocks of EfficientNetV2-B0 (Tan & Le, 2021), comprising only 1.39M parameters. Each branch extends from these layers, incorporating the sixth block of EfficientNetV2-B0. To enhance efficiency, BICEC is initialized with EfficientNetV2-B0 weights pretrained on ImageNet (Deng et al., 2009), enabling faster convergence. Since standard transfer learning requires matching architectures, BICEC Phase 1 (P1) ensures each branch ($Br$) plus shared layers ($S$) mirrors the pretrained EfficientNetV2-B0 network, allowing partitioning of pretrained weights ($Pw$) across shared ($Pw_0, \ldots, Pw_S$) and branch-specific layers ($Pw_S, \ldots, Pw_n$).

**Training.** P1 optimizes shared layers for subsequent branch scaling in Phase 2 (P2). Each branch is initialized with EfficientNetV2-B0 weights pre-trained

Table 1: BICEC functions.

| Ref | Function | Activation Condition |
|-----|----------|----------------------|
| M1 | Object Detection | Animates |
| M2 | Segmentation | People |
| M3 | Face Detection | Faces |
| M4 | Pose Detection | 3+ People |
| M5 | Action Recognition | Call, Text, Eat, Drink |
| M6 | Segmentation | Clothing Accessories |

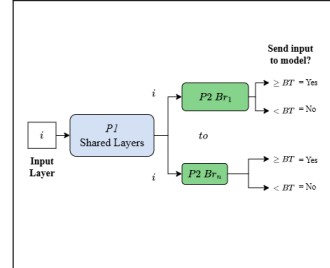

Figure 2: Example BICEC classifications.

on ImageNet and with its own input layer, output layer, and unique dataset (See Appendix A.1). The combined loss ($\sum_{i=1}^{n} L_i$) updates shared layers, mitigating conflicting updates. Each branch updates its own layers independently. Training continues until branch accuracy plateaus, selecting the configuration with the highest average branch accuracy. Final weights are saved for P2.

## 2.2 TRAINING PHASE 2: BRANCH SCALING

**Design.** In P2, BICEC retains its structure but reduces branch sizes to decrease parameters and FLOPs while maintaining close-to P1 performance. We re-scale each branch to a fraction of its P1 width and depth until a minimal P2 branch scale is found that achieves close-to it's P1 counterpart. This process, supported by *Uniform Element Selection* (UES) proposed by Xu et al. (2023), involves uniformly selecting weights from the source branch at each layer by sampling evenly spaced indices along each dimension of the weight tensor. Given a source tensor $W_t$ of shape $(t_1, t_2, \ldots, t_n)$ and a

target tensor $W_s$ of shape $(s_1, s_2, \ldots, s_n)$, with $s_i \leq t_i$, the selection extracts $s_i$ evenly spaced slices from $t_i$ at each layer. These weights are then initialized in the target layers, preserving structural consistency and enabling partial knowledge transfer.

**Training.** The complexity of features learned by each branch varies due to several factors, including the number of objects associated with a single activation condition, object sizes, the abstractness of activation conditions, and feature variation. These complexities make it challenging to predict optimal branch scales. BICEC refines scales iteratively. A set of scales $R$ is defined from manually specified minimum ($L$) to maximum ($U$) thresholds:

$$\text{Scales} = \left\{ L + \frac{U - L}{s - 1} \cdot (n - 1) \mid n \in \{1, 2, \ldots, s\} \right\}. \tag{1}$$

Shared layers, frozen with P1 weights, support branches initialized at scale $L$ with weights gained from UES. An allowable accuracy drop $a$ is set, iterating through $R$ until $Br_R = \min(Br_{P1})$, subject to $A_R + a \geq A_{P1}$, where $A_R$ and $A_{P1}$ denote reduced and full-sized branch accuracies. If no scale meets this criterion, the best-performing $A_R$ is selected. By P2 completion, the network satisfies $\#(Lb + S) \leq TP \leq \#(Ub + S)$, where $\#(\cdot)$ is number of parameters, $TP$ is total parameters, $b$ is branch count, and $S$ denotes shared layers.

Branches are scaled using the same unique datasets mentioned in P1. Weights for all tested s are stored, retaining the top-performing epoch's weights to ensure re-achieving P2 branch performance is not challenging. Freezing shared layers ensures consistent feature extraction across branches and reduces the complexity of the P2 training scenario leading to faster convergence.

## 2.3 ADDITIONAL FEATURES

**Branch Removal.** BICEC allows branch removal without retraining, reducing computational cost while preserving shared layers and remaining branches. Removed branches can be stored and re-integrated as needed.

**Branch Addition.** BICEC is designed to accommodate the addition of new vision system models by extending a new branch from its shared layers. The process for initializing the new branch and adapting the shared layers follows a two-phase alternating strategy: (1) *Step* ($St$) – the new branch is trained for $N$ epochs while the shared layers remain frozen; (2) *Pull* ($Pu$) – the new branch is frozen, and shared layers update for $N$ epochs using accumulated losses from all branches. This alternation continues as $St_{1,2} \rightarrow Pu_{1,2} \rightarrow St_{3,4} \rightarrow Pu_{3,4} \rightarrow \ldots$ with a reduced learning rate in $Pu$ to prevent instability. The new branch then enters Phase 2 ($P2$, Section 2.2). The updated network size follows $TP_{new} = TP_{old} + \#(L) \geq Br_{new} \leq \#(U)$, where $TP_{new}$ is total parameters with the added branch, $TP_{old}$ is total parameters before the branch was added, and $\#(L), \#(U)$ is the parameter count of the lower and upper branch scale.

During branch addition, BICEC restricts prior branches to processing inputs only during the pull phase to accumulate losses for shared layer adaptation. In contrast, during the step phase and scaling of the new branch, prior branches remain computationally inactive and receive no inputs. This design ensures efficient adaptation without incurring the expense of full network re-training.

**Binary Threshold Adjustment.** Each branch produces a binary score between 0 and 1, with a default classification threshold of 0.5. If the branch outputs a score exceeding this threshold, the model associated with the branch is activated. We can lower the threshold for each branch increasing relevant model activations at the cost of increasing irrelevant model activations.

## 3 EXPERIMENTS

**Training Details.** We use EfficientNetV2-B0 as the backbone of BICEC, providing a lightweight architecture with high throughput and low GFLOPs, reducing both response time bottlenecks and energy consumption (Tan & Le, 2021). Training and inference are conducted on an RTX 3070 (8GB). The training setup includes a learning rate of 0.0001 for shared layer creation, 0.00005 for shared layer adaptation during branch addition, and 0.0005 for the remaining training. Other parameters include batch size = 16, clip norm = 0.0001, Binary Crossentropy loss, and the Adam

optimizer (Kingma & Ba, 2014). Training is conducted for up to 20 epochs for shared layer creation, 8 epochs per scale, and 5 steps with 5 pulls of two epoches for branch addition.

**Training Datasets.** Each branch in BICEC has its own associated dataset consisting of 3156 positive (activation required) and negative (no activation required) examples. Input Relevance Tasks (IRTs) are used to train and evaluate BICEC, containing relevant and irrelevant inputs for each system model. We use two IRT versions: *IRT-Baseline (IRT-B)*, a demonstration system for BICEC experiments, and *IRT-Extended (IRT-E)*, which evaluates the impact of branch addition. Appendix A.1 details dataset characteristics and activation conditions.

**Metrics – Accuracy.** We first measure binary accuracy for individual branches and their combined outputs. Additionally, we reuse Correct Model Activation (CMA) from Burton-Barr et al. (2024), which quantifies how often BICEC correctly classifies an input as model relevant. Conversely, Incorrect Model Activation (IMA) measures the frequency of incorrect classifications.

**Metrics – Cost.** The 5000-image *COCO validation* set contains less person-oriented data, making it less relevant to the activation conditions of IRT models outlined in Appendix A.1. The *Movies* dataset consists of 3000 temporally ordered frames from three Jason Bourne films in the Movie Identification Dataset (Kaggle, 2023b). Lastly, the 5658-image *Y-VLOG* dataset is created by sampling every 24th frame from five YouTube travel video logs. The cost reduction is the sum of the activation cost of each model, $C(m_i)$, multiplied by the number of times the model is activated, $N(m_i)$, divided by the total number of images in the dataset, $IS$, for all models $m_i$ from 1 to $M$.

Energy and inference time reductions follow the same calculation, with energy reduction requiring an estimation of per-inference energy consumption. This is approximated by dividing the model's total FLOPS by the ratio of the maximum FLOPS per second to the maximum wattage per second. Given that an RTX 3070 (8GB) has a theoretical performance of 20.31 TFLOPs at a maximum power draw of 220W, this results in approximately 92.3 GFLOPS per watt. Appendix A.2 provides model information for IRT-B and IRT-E, including inference and energy cost of each model.

## 3.1 NETWORK ANALYSIS

**Scaling.** Scaling significantly reduced BICEC's size while revealing varying branch size requirements. The reduction depended on $a$, where larger values allowed greater compression. A small $a$ maintained performance with moderate reductions (Table 2); for instance, $a = 0.005$ reduced model size by 33.12%. Increasing $a$ to 0.020 reduced the model by 87.84% while incurring only a 1.8% accuracy drop. Additionally, FLOPs decreased by 54.42%. Figure 3 visualizes accuracy changes per branch at $a = 0.020$. With similar performance at a fraction of the size, we selected BICEC at $a = 0.020$ for remaining experimentation. Examples of BICEC's classifications are in Figure 2.

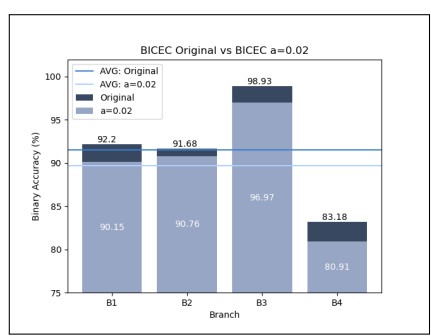

Figure 3: Branch accuracy and scaling.

Table 2: Scaling results for $U = 9$. (*) indicates the maximum scale did not reach the target.

| $a$ | $Br_1$ | $Br_2$ | $Br_3$ | $Br_4$ | Model Size | GFLOPs | Acc (%) |
|---|---|---|---|---|---|---|---|
| Orig. | 1.0 | 1.0 | 1.0 | 1.0 | 19.50M | 2.591 | 91.50 |
| 0.005 | 0.9* | 0.9 | 0.9* | 0.3 | 13.03M | 2.055 | 90.90 |
| 0.010 | 0.9 | 0.7 | 0.7 | 0.2 | 8.39M | 1.706 | 90.54 |
| 0.015 | 0.9 | 0.6 | 0.3 | 0.3 | 6.55M | 1.520 | 90.02 |
| 0.020 | 0.4 | 0.4 | 0.3 | 0.2 | 2.47M | 1.181 | 89.70 |

**Binary Threshold Adjustment.** We see the impact of binary threshold adjustment in Table 4. As the binary threshold decreases from 0.5 to 0.0625, CMA increases from 86.43% to 94.55%, indicating improved model activation for relevant inputs. However, this comes at the cost of higher IMA, which rises from 8.16% to 27.42%. Model accuracy correspondingly declines from 89.70%

to 84.01%, illustrating a trade-off between activation sensitivity and prediction reliability. Lower thresholds enhance CMA but increase false activations and reduce accuracy.

**Branch Addition.**  Contrary to expectations, Table 3 shows that adding branches improved accuracy, potentially due to careful adaptation. A lower learning rate may help shared layers generalize better, mitigating overfitting (Ruder, 2017). Scaling reduced branch parameters from 4.60M to 99.2K ($Br_5$) and 10.2K ($Br_6$) while preserving accuracy within $a = 0.020$, minimizing the increase in total BICEC GFLOPs. New branches performed well in both accuracy and CMA, though performance will vary based on dataset and activation conditions. In regards to training, no re-tuning of prior branches was required during the shared layer adaption and subsequent new branch scaling phase.

Figure 4: Binary Threshold (BT) Adjustment for IRT-B

| BT | CMA | IMA | Accuracy |
|---|---|---|---|
| 0.5 | 86.43% | 8.16% | 89.70% |
| 0.25 | 90.98% | 15.15% | 88.45% |
| 0.125 | 93.26% | 21.06% | 86.60% |
| 0.0625 | 94.55% | 27.42% | 84.01% |

Table 3: Extending the network to create the IRT-E version of BICEC. Accuracy (A), Network (Net), Branch (Br), Branch Post-Scaling (Br-S).

| | Scale | $\Delta$Params | $\Delta$GFLOPs | Net A | Br A | Br-S A | Net CMA | Br CMA |
|---|---|---|---|---|---|---|---|---|
| IRT-E T5 | 0.2 | +99.2K | +0.01 | +0.12% | 90.76% | 89.55% | -0.22% | 90.00% |
| IRT-E T6 | 0.1 | +10.2K | +0.003 | +0.66% | 99.39% | 99.09% | +0.79% | 99.39% |

## 3.2 Cost Analysis

Table 4 highlights the average inference time and energy cost reductions across models in IRT-B and IRT-E. On average, total energy consumption dropped by 52.1% (from 4.34W to 2.08W), while total inference time decreased by 54.7% (from 141.0ms to 63.8ms). Variability across datasets (COCO-Val, Movie, and Y-VLOG) is linked to task characteristics and differing relevance of models to parsed data. For example, we can observe how COCO-Val's less person-oriented distribution reduced model applicability compared to Movie and Y-VLOG.

Table 4: Energy and inference costs for each model associated with IRT-B and IRT-E. See Appendix A.1 or Appendix A.2 for more information on model costs and model functions.

| | Inference (ms) | | | | | | Energy (W) | | | | | |
|---|---|---|---|---|---|---|---|---|---|---|---|---|
| | M1 | M2 | M3 | M4 | M5(E) | M6(E) | M1 | M2 | M3 | M4 | M5(E) | M6(E) |
| **Standard** | 17 | 33 | 13 | 12 | 46 | 20 | 1.18 | 0.86 | 0.45 | 0.88 | 0.38 | 0.67 |
| COCO-Val | **9.1** | **17.1** | **3.6** | 4.1 | **8.5** | 6.0 | **0.63** | **0.45** | **0.13** | 0.30 | **0.07** | 0.20 |
| Movie | 10.5 | 20.0 | 9.1 | **2.3** | 24.6 | 7.4 | 0.73 | 0.52 | 0.32 | **0.17** | 0.20 | 0.25 |
| Y-VLOG | 10.9 | 22.6 | 7.5 | 4.3 | 17.9 | **5.9** | 0.76 | 0.59 | 0.26 | 0.31 | 0.15 | **0.20** |

Cost reductions also varied between system models with some activation conditions having higher frequency (e.g. M1 and M2) than others (e.g, M3 and M4). For IRT-B, BICEC contributed an energy cost of just 0.0127W per inference, 0.74% of the system's average wattage. In IRT-E, the energy cost rose marginally to 0.0129W (0.62% of system wattage). Notably, BICEC's impact on inference time was more pronounced, accounting for 10.1ms (25.0% of inference time) for IRT-B and 11.2ms (17.6%) for IRT-E. IRT-E particularly shows how BICEC's relative costs can reduce for larger AI systems with a greater roster of models.

## 3.3 Comparison with SICEC

BICEC showed advantages over a previously existing attachable vision system controller named SICEC (Burton-Barr et al., 2024). Developing on SICEC's single-label activation controller, BICEC achieves higher accuracy with a smaller model size for IRT-B. Despite initial under-performance in CMA, threshold adjustments resolved this issue (Appendix A.3). A key improvement is decision space growth: BICEC scales linearly, as SICEC can grow exponentially as independent models are added (Appendix A.4). Although SICEC can reduce this growth through model dependencies, where some models are dependent on the activation of other models, such conditions are not always

guaranteed. SICEC's decision space growth creates difficulties in dataset curation where each combination of activations need to be represented. Additionally, SICEC did not cover scenarios where models are added or removed from the attached system. We argue that BICEC better accommodates computer vision systems, adapting to system changes and attaining improved accuracy and CMA.

## 4 RELATED WORK

**Dynamic Vision Systems**. While dynamic inference control is well-established for single-model applications (Wang et al., 2020; Laskaridis et al., 2021; Ahn et al., 2019; Zhang et al., 2021), control across multiple models remains relatively less explored and often requires system re-training to incorporate the proposed method. For example, AdaMTL (Neseem et al., 2023) co-trains task-aware policy networks to determine block activation within sub-networks for each input alongside multi-task networks. Additionally, AdaMV-MoE (Chen et al., 2023) achieves dynamic input adaptiveness through sparse mixture of expert selection via a task-dependent router networks; however, expert creation and selection is tightly integrated with the training of the multi-task network. In contrast, SICEC (Burton-Barr et al., 2024) provides attachable activation control for vision system models by using single-label image classification to declare input-relevant models for activation. Unfortunately, SICEC's single-label architecture is the primary cause of issues discussed in Section 3.3. In extension aforementioned methods overlook changes to a system's models during its lifetime, where incorporating adaptiveness could reduce retraining costs and maintain performance.

**Methodological Inspirations**. EfficientNetv2 (Tan & Le, 2019; 2021) employs a range of optimization techniques including fused MBConv layers and compound scaling to produce computationally efficient models that can help reduce system bottlenecks for resource-conscious applications. B-CNN (Zhu & Bain, 2017) demonstrated that branching can enhance performance in cases where class difficulty is inconsistent, resulting in an accuracy improvement of 1.50–6.59% compared to baseline models on CIFAR-100. Similarly, BRNet (Gupta et al., 2022) highlighted that branching can increase the diversity of features learned from the input. Transfer learning (Weiss et al., 2016; Zhuang et al., 2020) and UES (Xu et al., 2023) were particularly important for more efficient training in this work, accelerating convergence by recycling previously learned feature representations.

## 5 DISCUSSION

**Benefits to Sustainable AI Systems**. BICEC contributes to the development of sustainable AI systems by addressing critical challenges in resource consumption and computational efficiency. By intelligently controlling model activation based on input characteristics, BICEC reduces unnecessary model activations, leading to significant reductions in inference time and energy consumption. While BICEC focuses on intelligent activation control of vision systems, we support that modulation and tighter control of AI models can reduce computation and resulting energy costs. By preventing unnecessary energy consumption, BICEC contributes to reducing the environmental impact of AI systems, thereby enhancing their sustainability.

**Limitations**. While BICEC emphasizes the complete prevention of model activation, other approaches have explored more granular control strategies. For example, sparse selection of experts (Chen et al., 2023), selective activation of model blocks (Neseem et al., 2023), and input-dependent model-size selection (Burton-Barr et al., 2024) may offer additional computational efficiencies and more precise input-model alignment, which remain unexplored in BICEC. Additionally, further testing of activation condition complexity would be ideal, for example a future application might require BICEC branches to understand a broader range of input features and not just a select few. Finally, BICEC was designed as an example of vision system control, however, future research should consider multi-modal systems.

**Conclusions**. BICEC is a non-invasive, attachable solution for vision systems that requires no fine-tuning of pre-trained models within the system. BICEC uniquely enables adaptation to system changes by supporting branch removal without training and branch addition with reduced training. In extension, BICEC's results show significantly lower average inference time (-54.7%) and energy consumption (-52.1%) in attached system models. We claim that BICEC demonstrates how tighter regulation of model activation in AI systems can reduce computational overhead, optimize energy consumption, and better align the attached system with the goals of sustainable AI.

## 6 ACKNOWLEDGEMENT

This research is supported by funding allocation to Basura Fernando by the Agency for Science, Technology, and Research (A*STAR) under its SERC Central Research Fund (CRF), as well as its Centre for Frontier AI Research (CFAR).

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

# A APPENDIX

## A.1 DATASET DETAILS

Table 5: Datasets and models for IRT-B and IRT-E.

| Task | Ref | Model | Function | Activation Condition | Origin | Train-size | Val-size |
|---|---|---|---|---|---|---|---|
| IRT-B, IRT-E | M1 | YOLOv5l | Object Detection | Animates | COCO (Lin et al., 2014) | 5652 | 660 |
| IRT-B, IRT-E | M2 | SegFormer-B3 | Segmentation | People | COCO | 5652 | 660 |
| IRT-B, IRT-E | M3 | YOLOv5l-Face | Face Detection | Faces | FDDB(Jain & Learned-Miller, 2010) | 5652 | 660 |
| IRT-B, IRT-E | M4 | YOLOv8m-Pose | Pose Detection | 3+ People | COCO | 5652 | 660 |
| IRT-E | M5 | ViT-B16_x224 | Action Recognition | Call, Text, Eat, Drink | HAR (Kaggle, 2022) | 5652 | 660 |
| IRT-E | M6 | SegFormer-B2 | Segmentation | Accessories | FA-Grouped (Kaggle, 2023a) | 5652 | 660 |

## A.2 MODEL DETAILS

Table 6: IRT-B and IRT-E model details. input size, parameters, and flops(G) reported in papers. Energy (W) is calculated as mentioned in the cost-metrics and Inference (ms) details the time taken to run a single input through the model on our RTX 3070.

| Ref | Model | Input Size | Parameters (M) | Inference (ms) | Flops(G) | Energy (W) |
|---|---|---|---|---|---|---|
| M1 | YOLOv5l (Jocher, 2020) | 640x640 | 46.5 | 17 | 109.1 | 1.18 |
| M2 | SegFormer-B3 (Xie et al., 2021) | 512x512 | 47.3 | 33 | 79.0 | 0.86 |
| M3 | YOLOv5l-face (Qi et al., 2022) | 640x640 | 46.6 | 13 | 41.6 | 0.45 |
| M4 | YOLOv8m-Pose(Ultralytics, 2024) | 640x640 | 11.6 | 12 | 81.0 | 0.88 |
| M5 | ViT-B16(Dosovitskiy et al., 2020) | 224x224 | 86.9 | 46 | 35.2 | 0.38 |
| M6 | SegFormer-B2 (Xie et al., 2021) | 512x512 | 28.0 | 20 | 62.0 | 0.67 |

## A.3 SICEC AND BICEC COMPARISON: PERFORMANCE

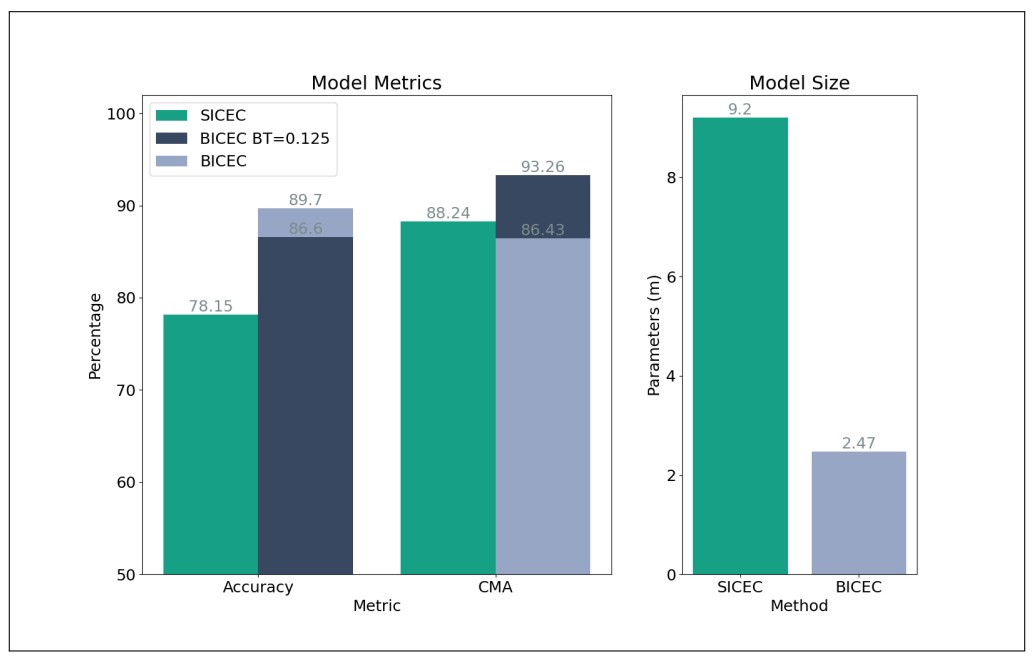

Figure 5: SICEC IRT-B compared with BICEC IRT-B.

## A.4 SICEC AND BICEC COMPARISON: DECISION SPACE

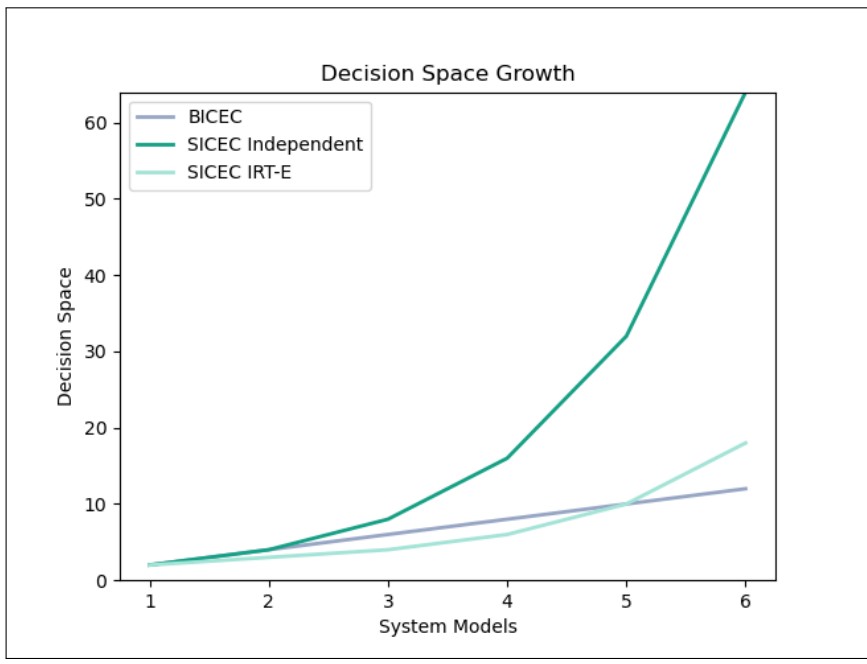

Figure 6: SICEC decision space growth compared to BICEC.

