# OpenReview forum: "BICEC: Attachable Classification-Based Intelligent Control for Sustainable Computer Vision Systems"
_ICLR.cc/2025/Workshop/MCDC — MCDC @ ICLR 2025_

### Official Review · Reviewer_TvJj · 2025-02-20

**Rating:** 7
**Confidence:** 5
**Fit:** 4

**Summary:**

The paper introduces BICEC (Branched Image Classification Evaluative Controller), an intelligent control mechanism designed to optimize the activation of computer vision models within multi-model AI systems. The key goal is to enhance energy efficiency and inference speed by reducing unnecessary model activations. The authors propose an attachable, classification-based approach using a branched EfficientNetV2 architecture. The system can be integrated into existing AI models without retraining them and achieves substantial reductions in computational cost.

The core contributions of this paper are:

1. A lightweight branched neural network based on EfficientNetV2 that selectively activates relevant models.
2. A two-phase training process that balances model efficiency and activation accuracy.
3. An attachable design, allowing seamless integration with existing vision pipelines.
4. Demonstrated improvements in energy efficiency and inference speed through experimental results.
5. The authors compare BICEC with an existing attachable control system, SICEC, highlighting superior accuracy, efficiency, and adaptability.

**Reason For Giving A Higher Score:**

The higher score is justified because BICEC presents an innovative and practical approach to optimizing model activation in multi-model vision systems, significantly improving computational efficiency and sustainability. The experimental results demonstrate notable reductions in inference time (-54.7%) and energy consumption (-52.1%), highlighting its effectiveness in reducing computational overhead without compromising accuracy. Its scalability and flexibility, particularly the ability to add or remove models without retraining, make it a highly adaptable solution for evolving AI systems.

Also, stated the reasons in the pros of the paper

**Reason For Giving A Lower Score:**

NA

**Strengths And Weaknesses:**

Pros of the paper:
1. The research is very relevant in the context of sustainable AI, and it addresses the increasing computational and energy demands of deep learning models. By regulating model activation dynamically, BICEC aligns with recent efforts to reduce AI's carbon footprint and optimize resource utilization.

2. The paper also presents a novel approach by introducing a branched classification-based mechanism that enables an intelligent activation control. Contrast to the existing approaches, BICEC

     a. Uses a structured two-phase training process to optimize efficiency.\
     b. Allows seamless branch addition and removal without requiring system retraining.\
     c. Leverages transfer learning and Uniform Element Selection (UES) for efficient weight adaptation.

3. The authors have also provided a thorough evaluation using multiple datasets (COCO, Movie, Y-VLOG) and various metrics. The energy and inference time reduction (52.1% and 54.7%, respectively) validate BICEC's effectiveness in improving computational efficiency.

     a. Performance Metrics: Accuracy, Correct Model Activation (CMA), and Incorrect Model Activation (IMA).\
     b. Computational Efficiency Metrics: FLOPs reduction, inference speed, and energy consumption.\
     c.  Comparison with SICEC: BICEC shows better scalability and activation accuracy while reducing decision space complexity.


4. BICEC is designed to be integrated into real-world vision systems without requiring modifications to existing models. This plug-and-play nature makes it applicable to diverse AI systems where multiple vision models operate in parallel.

Areas for Improvement:

1. While the paper compares BICEC with SICEC, it does not benchmark against dynamic model selection techniques such as Mixture of Experts (MoE) or adaptive multi-task learning approaches (AdaMTL, AdaMV-MoE). Including a comparison with these systems would provide a more comprehensive analysis of BICEC’s advantages and trade-offs.

2. The activation conditions defined in the paper(e.g., object detection, segmentation, action recognition) are very limited. They mostly focus on high-level vision tasks. However, many real-world applications require finer-grained decisions, such as:

     a. Scene-dependent activation (e.g., detecting road signs in autonomous driving).\
     b. Context-aware model switching in multi-modal systems.\
     Exploring more complex decision boundaries for model activation would strengthen the generalizability of BICEC.

3. The paper also discussed the impact of binary threshold adjustment on CMA and IMA but does not explore adaptive thresholding mechanisms that dynamically adjust thresholds based on input uncertainty. A future improvement could involve self-tuning thresholds that optimize trade-offs between false activations and missed activations.

4. The paper has estimated the energy consumption based on theoretical FLOPs-to-Watt conversion for an RTX 3070 GPU. However, actual energy usage can vary due to:\
      a. Memory bandwidth limitations.\
      b. GPU power management states.\
      c. System overhead (e.g., data loading time, I/O operations).\
      Using a more precise energy measurement tool (e.g., NVIDIA's NVML API) would provide a more accurate assessment of BICEC’s power savings.

5. The datasets used (COCO, Movie, Y-VLOG) are well-known but limited in domain coverage. Introducing more diverse datasets (e.g., medical imaging, autonomous driving) would demonstrate BICEC’s generalizability.

**Suggestions:**

1. While the paper compares BICEC with SICEC, it would be beneficial to benchmark against Mixture of Experts (MoE) approaches and adaptive multi-task learning (AdaMTL, AdaMV-MoE).  Adding a quantitative comparison (e.g., efficiency gains, accuracy trade-offs) against dynamic model selection techniques would help establish BICEC's relative strengths.

2. Expanding BICEC to handle context-dependent activations, such as dynamic scene changes or temporal dependencies, would increase its real-world applicability.

3. Implementing self-adjusting thresholds based on confidence scores or input uncertainty could improve Correct Model Activation (CMA) while reducing Incorrect Model Activation (IMA). The study explores fixed binary threshold values but does not investigate adaptive thresholding mechanisms.

4. The paper focuses solely on computer vision models. Extending BICEC to multi-modal systems (e.g., vision + language, vision + audio) could broaden its impact and make it relevant for LLMs and multi-modal AI applications.

---

### Official Review · Reviewer_ZPcd · 2025-02-26

**Rating:** 7
**Confidence:** 4
**Fit:** 4

**Summary:**

The paper proposes a control system for vision tasks in the research vain of SICEC (Burton-Barr et al, 2024). The proposed methodology aims for a more computationally efficient model selection in terms of time and energy cost. The main contribution is a process for branch creation and scaling, emphasizing weights transfers.

**Reason For Giving A Higher Score:**

It is good paper that focuses on the combination & modularization of existing models via an intelligent control. Whilst specific to vision, it may be extendable to other domains. It fits well within the theme of the workshop and the results are encouraging.

**Reason For Giving A Lower Score:**

The comparison versus existing approaches can be expanded (in addition to discussed). Furthermore, there is some lack of clarity in what parameters are trained and when: further clarification would be beneficial.

**Strengths And Weaknesses:**

The authors provide a clear explanation of the proposed methodology (if somewhat high-level) and the motivation for the work. The comparison versus existing methods can be expanded but overall the results are supportive of authors' claim of improving the efficency (time and energy). The claims of an attachable and adaptable system are only partially substantiated: more results on these two aspects would be great.

**Suggestions:**

The main suggestions are:
(a) Expand on the use of EfficientNetV2-B0, which seems to be integral to their proposed construct; and
(b) Focus on substantiating the claim of adaptability (end of Section 1), which is merely discussed but not tested.

---

### Official Review · Reviewer_msSw · 2025-03-03

**Rating:** 4
**Confidence:** 4
**Fit:** 3

**Summary:**

The paper presents BICEC, a method to select which model(s) to apply for a given input image among a pool of available models, to perform several tasks. BICEC is built on using a pretrained EfficientNet-v2 backbone, and has minimal cost compared to the models themselves.  It is trained on different phases: First, the last block of EfficientNet-v2 is trained to predict whether to use each of the available models in the pool (using binary cross entropy). Then, the size of the last block is reduced to reduce the total cost of BICEC while trying to keep the classification accuracy. During inference, all the models for which the BICEC score of a given input surpases a predefined threshold (hparam), are activated. The paper compares BICEC against SICEC, a similar approach published in 2024. BICEC offers more flexibility than SICEC, since it allows for cheap brach removal and addition.

**Reason For Giving A Higher Score:**

I would absolutely need to see a proper evaluation against SICEC and other benchmarks preferably, that takes total cost into account. See also the other weaknesses that I mentioned that would need to be addressed for a higher score.

**Reason For Giving A Lower Score:**

The proposed method seems to have some benefits that the baseline does not. Thus, I'm not giving a lower score.

**Strengths And Weaknesses:**

**Strengths**
- The method performs better than the baseline (SICEC) with a smaller model size (9.2M vs 2.47M params), after tuning some hyperparameters. Note however, that there are some caveats with this comparison (see weaknesses regarding cost below).
- The proposed method allows add support for additional models in the future (i.e. see Branch addition experiments in section 3.1).


**Weaknesses**
- The (down)scaling of the last block of the EfficientNet-v2, that is copied and used to output a binary classification on whether to use the different models is not entirely justified. Table 2 shows how the GFLOPs of BICEC are reduced, but this doesn't show the total cost relative to running BICEC plus the actual models. Table 6 contains the GFLOPs of the different models and they are in the range of [35.2, 109.1] GFLOPs, thus reducing the cost of BICEC from 2.6 to 2.0 or even 1.2 makes almost no difference in practice: -3.8% total GFLOPs reduction being super optimistic, assuming only the cheapest model is always used; -1.3% GFLOPs if only the most expensive model is selected; or ~1% total GFLOPs if 2 models are selected per input with uniform frequency.
- The latter makes the comparison with SICEC a bit incomplete: could BICEC be better than SICEC simply because it activates more models per input?
- The paper only compares against a single baseline (SICEC), but it is not clear if the evaluation was done under comparable conditions (same EfficientNet-v2 architecture? same training data & evaluation benchmarks?). In terms of alternative methods, the authors cite a decent amount of different alternatives in Section 4, but they only compare against SICEC.
- The architecture used for BICEC is quite old (in realtive terms for computer vision research). It uses an EfficientNet-v2, from 2021. It's not clear why this architecture is preferred instead of more modern alternatives (e.g. small ResNets or Transformers). I guess that comparing against SICEC is the obvious anwer, but this reviewer keeps wondering if the approach works with other architectures and waht the performance would be.
- The abstract reads "BICEC adapts to existing vision systems without requiring system retraining". When one reads this one might get the impression that the proposed method is "zero-shot", but it actually requires the tuning of the EfficientNet backbone used to build BICEC.

**Suggestions:**

Most imporantly, please make the appropriate changes to address the weaknesses that I mentioned.


Other non-critical but nice improvements to the text:
- Address the fine-tuning requirements in the abstract, it's a bit misleading as it is.
- Please, fix scientific notation. I assume that the "learning rate of $1e^{-4}$" (line 160) actually means $10^{-4} = 0.0001$ and not $e^{-4} = (2.7182...)^{-4} \approx 0.0183156$.
- In lines 124-125: $Lb$ and $Ub$ mean $L \cdot b$ and $U \cdot b$ respectively. However $Sl$ is a single symbol, and does not mean $S \cdot l$. This is confusing to the reader. Maybe use $\text{Sl}$, or simply $S$ for the latter?
- The symbol $R$ in line 114-115 is not used anywhere. It refers to the "set of scales", but then when this set is defined in Eq. (1) the paper uses "Scales = {" rathern than "$R$ = {". So, it is a completely unnecessary symbol.

---

### Official Review · Reviewer_zmHC · 2025-03-04

**Rating:** 6
**Confidence:** 3
**Fit:** 4

**Summary:**

The proposed paper introduces **BICEC**, an attachable classification-based intelligent controller designed to optimize model selection and use in computer vision systems. The proposed key contribution lies in its *branched neural network architecture* derived from a pre-trained EfficientNetV2 backbone, which employs *shared layers coupled with model-specific branches* that output binary decisions regarding model activation. The system operates in a two-phase training process : the first phase establishes a *base configuration using transfer learning*, and the second phase *iteratively scales each branch via Uniform Element Selection (UES)* to reduce parameters and FLOPs while preserving near-optimal activation performance. BICEC’s modular design supports non-invasive integration with existing systems, allowing branch removal or addition without necessitating full retraining, thereby addressing dynamic system changes. Experimental evaluations on multiple datasets demonstrate *reductions in inference time* (\~55%) and *energy consumption* (\~52%), while *maintaining a good correct model activation accuracy*. Overall, the work contributes to show that tighter, data-driven regulation of model activation can enhance both computational efficiency and sustainability in AI systems.

**Reason For Giving A Higher Score:**

The paper presents an interesting and practical approach to enhancing energy efficiency and inference speed in computer vision systems. The architecture and training methodology, along with comprehensive internal evaluations, offer a compelling case for the benefits of attachable control. The flexibility to add or remove branches without full retraining is particularly appealing for sustainable applications.

**Reason For Giving A Lower Score:**

The paper’s comparative analysis is limited in scope, with most comparisons confined to a single baseline (SICEC) and primarily detailed in the appendix. This narrow focus raises questions about how BICEC stacks up against a broader range of alternative methods. Additionally, the reliance on binary activation and sensitivity to parameter tuning may present challenges in more diverse or real world applications.

**Strengths And Weaknesses:**

**Strengths :**
- *Architecture and training strategy :* BICEC’s design is novel for such computer vision use, employing a two-phase training process that enables attachable and adaptable control without requiring full system retraining.
- *Efficiency gains :* The method demonstrates significant reductions in energy consumption and inference time, which is crucial for sustainable AI and resource-constrained environments.
- *Adaptability :* The support for branch removal and addition offers flexibility to handle evolving system models with minimal reconfiguration effort.
- *Comprehensive internal analysis of the method :* Detailed experiments and network analyses (e.g., branch scaling, binary threshold adjustment, and cost reduction evaluations).

**Weaknesses :**
- *Limited comparative analysis :* The paper primarily compares BICEC with SICEC, and this comparison is presented only briefly and largely relegated to the appendix. There is a lack of broader contextualization with alternative methods such as dynamic routing or multi-task learning frameworks.
- *Overemphasis on self-comparison :* Most experimental results focus on demonstrating improvements relative to BICEC’s internal baselines rather than contrasting its performance with a wider array of existing architectures.
- *Simplified activation mechanism :* The use of a binary classification approach for model activation may not capture more nuanced or multi-label activation scenarios that could be beneficial in complex environments.
- *Parameter sensitivity and robustness :* The system’s performance appears sensitive to key parameters (e.g., accuracy drop thresholds, binary threshold adjustments), and the paper could benefit from a more extensive robustness analysis under varied and noisy input conditions.

**Suggestions:**

- *Expand comparative analysis :* Consider including additional comparisons with other dynamic activation control strategies, such as models that employ dynamic routing, sparse expert selection, or multi-task learning frameworks. A more comprehensive evaluation against diverse baselines would strengthen the claims of improved efficiency and adaptability.
- *Enhance discussion on activation mechanisms :* Explore the potential of extending beyond binary classification for model activation. Discussing or experimenting with multi-label or probabilistic approaches might yield insights into handling more complex input conditions.
- *Robustness and sensitivity analysis :* Incorporate additional experiments that test the system’s sensitivity to parameter variations and evaluate its performance under noisy or unexpected input conditions. This could help to better understand and mitigate potential limitations in real-world scenarios.

---

### Decision · Program_Chairs · 2025-03-06

**Decision:**

Accept

**Comment:**

This work investigates modularity of models based on speed and energy efficiency, this modular aspect is very relevant to that workshop. Some reviewers have noted possible improvements, that we encourage the authors to attend to. However, the majority recommended the paper to be accepted, and thus we're please to accept this paper to the workshop.